# Histamine Intolerance in Children: A Narrative Review

**DOI:** 10.3390/nu13051486

**Published:** 2021-04-28

**Authors:** Wojciech Nazar, Katarzyna Plata-Nazar, Katarzyna Sznurkowska, Agnieszka Szlagatys-Sidorkiewicz

**Affiliations:** 1Faculty of Medicine, Medical University of Gdańsk, Marii Skłodowskiej-Curie 3a, 80-210 Gdańsk, Poland; 2Department of Paediatrics, Gastroenterology, Allergology and Paediatric Nutrition, Faculty of Medicine, Medical University of Gdańsk, Nowe Ogrody 1-6, 80-803 Gdańsk, Poland; knazar@gumed.edu.pl (K.P.-N.); katarzyna.sznurkowska@gumed.edu.pl (K.S.); agnieszka.szlagatys-sidorkiewicz@gumed.edu.pl (A.S.-S.)

**Keywords:** children, histamine intolerance, nutrition, diagnostic algorithm, epidemiology

## Abstract

Histamine intolerance is defined as a disequilibrium of accumulated histamine and the capacity for histamine degradation. This clinical term addresses a non-immunologically mediated pathology when histamine ingested with food is not particularly high, however its degradation is decreased. This paper aims to provide a narrative review on etiopathology, epidemiology, possible diagnostic algorithms and diagnostic challenges of histamine intolerance in children. The clinical picture of histamine intolerance in children is similar to that observed in adults apart from male predominance found in paediatric patients. Both in children and adults, a histamine-reduced diet is typically the treatment of choice. Diamine oxidase supplementation offers another treatment option. There is no symptom or test pathognomonic for histamine intolerance. Nevertheless, manifestations of chronic gastrointestinal symptoms, measurements of diamine oxidase deficits, positive results of histamine provocation tests and improvement in symptoms with histamine-reduced diet considerably increase the probability of histamine intolerance diagnosis. These factors have been included in the proposed diagnostic algorithm for histamine intolerance. In children histamine intolerance most likely co-occurs with allergies and bowel diseases, which creates an additional diagnostic challenge. As the evidence for children is poor further research is needed the determine epidemiology, validate diagnostic algorithms and establish possible treatment options regarding histamine intolerance.

## 1. Introduction

Histamine intolerance—HIT—is defined as a disequilibrium of accumulated histamine and the capacity for histamine degradation [1]. This clinical term addresses a non-immunologically mediated pathology when histamine ingested with food is not particularly high, but its degradation is decreased [1,2]. Thus, some individuals are unable to degrade ingested histamine properly, which subsequently causes sensitivity to normal or even low histamine levels in food. Research on this topic has significantly accelerated during the last decade and the awareness of histamine intolerance has been growing among researchers and in the general public [3]. However, to date, no review on HIT in children has been published. As there are only a few research papers on HIT in children, this review was also based on data from studies on adult cohorts. Nevertheless, HIT seems to be a highly relevant issue in clinical paediatrics. This paper aims to provide a narrative review on the etiopathology, epidemiology, possible diagnostic algorithms as well as diagnostic challenges of histamine intolerance in children. The differences between the paediatric and adult populations are also highlighted.

## 2. Histamine Metabolism

Histidine decarboxylase catalyses the production of histamine from histidine (Figure 1). Histamine is metabolised by two primary enzymes: diamino oxidase, DAO, and histamine-N-methyltransferase, HNMT [3,4]. HNMT is responsible for intracellular histamine metabolism, while DAO is a secretory protein that metabolises histamine extracellularly [5]. DAO has a higher expression than HNMT [3], being the primary barrier for intestinal histamine absorption [5]. 

Imidazole acetaldehyde produced by DAO is subsequently metabolised by aldehyde dehydrogenase, ADH, to imidazole acetic acid. Acetic acid is then degraded into 1-ribosylimidazole-4-acetic acid. In the second metabolic pathway, N-methylhistamine produced by HNMT is oxidised by monoamine oxidase B leading to N-methyl imidazole acetaldehyde that is further transformed into methyl-imidazole acetic acid by ADH. Histamine can also be metabolised to Nω-acetylhistamine by gut bacteria [4,6].

## 3. Histamine Content in Foods

Based on food lists created by Rosell-Camps et al. for children on histamine-reduced diets [7] and available scientific database [1,8,9], lists of foods to be allowed (Table 1) and avoided (Table 2) on histamine-reduced diets were created. Some fish (fresh and processed), as well as other highly processed and fermented foods (mainly some cheese) dominate the high histamine list, while the low histamine list is made up predominantly of fresh foods and foods with minimal processing.

## 4. Histamine Intolerance or Histamine Intoxication?

In HIT, histamine accumulation in intestinal mucosa and blood plasma is caused by decreased histamine degradation, resulting most commonly from low DAO activity. Additionally, tissue HNMT is inhibited by histamine metabolites [7]. Thus, HIT can occur even when food with moderate or even low amounts of histamine are ingested.

This should be differentiated from the situation when individuals presenting with normal histamine degradation enzyme activity ingest large quantities of histamine-rich food (mainly fish) and develop histamine-induced symptoms, which is classified as histamine intoxication [3].

Histamine acting on its H_1_–H_4_ receptors [10] is a common denominator in both pathologies, which results in the appearance of histamine-mediated symptoms [3]. 

The recent meta-analysis shows that the mean histamine content in foods causing histamine intoxication is about 1100 mg/kg (95% confidence interval 422–2900 mg/kg) and almost all intoxications (98%) result from fish consumption [11]. Legal limits for histamine content in food products (mainly fish) are set, in the majority, at between 100 and 400 mg/kg [12]. Thus, the histamine content of food causing intoxication is substantively higher than stated in food safety regulatory guidelines. 

For comparison, it appears like foods that can be allowed on a histamine-reduced diet contain less than 5 mg/kg of histamine (Table 2). On the other hand, most recommendations for histamine-reduced diet list foods with a wide range of histamine content, including some foods which may contain histamine concentrations near the legal limits for histamine content in foods. (Table 2) [12].

## 5. Primary (Genetic) Histamine Intolerance

HIT can be a genetic condition (Figure 2). In this case, it is caused by single nucleotide polymorphisms, SNPs, in the DAO gene. Expression of this gene results in altered protein production with lower enzymatic activity than usual. In the adult population, the most important SNPs in the DAO gene causing this deficit are rs10156191, rs1049742, rs2268999 and rs1049793 [3,13,14]. SNPs in the promoter region that decrease the DAO gene expression (rs2052129) have also been reported [13]. Similarly to the adult population, the DAO SNP rs1049793 is also associated with lower DAO activity in children and leads to higher serum histamine levels in paediatric patients with allergic rhinitis [15].

The relationship between the SNPs in the HNMT gene (N-methyltransferase C314T allele) and allergic diseases was investigated in children in several studies [15,16,17,18,19]. It appears that HNMT SNPs are associated with more severe courses of atopic dermatitis and allergic rhinitis while for asthma and bronchial hyperresponsiveness the results are contradictory.

## 6. Secondary Histamine Intolerance Aetiology

### 6.1. Disease-Induced Histamine Intolerance

DAO deficit was found in children undergoing cyclophosphamide administration in neuroblastoma paediatric patients [20], in celiac disease [21], in acute gastroenteritis [22] and during protracted diarrhoea [23]. In adults, decreased DAO excretion was reported in other diseases including chronic urticaria, viral hepatitis [1] and inflammatory bowel diseases [3]. These pathologies also occur in childhood [24,25,26] but insofar no evidence of DAO deficit in those conditions has been reported in children, although, as one report describes, histamine intolerance was diagnosed in a child with viral intestinal infection [27].

Thus, it appears like DAO deficiency may be caused by various factors such as the presence of a primary disease or cytostatic treatment. The common denominator is intestine damage. Similar primary factors cause secondary lactose deficiency described as reduced lactase activity [28]. Lactase, an enzyme responsible for lactose digestion, is produced, similarly to DAO, by intestinal cells. Also similarly to DAO deficiency, symptoms of lactose malabsorption, LM, include flatulence, diarrhoea, abdominal pain as well as systemic symptoms like headaches [29,30]. Moreover, the exclusion of the same dietary products may lead to an improvement in symptoms. For example, a cheese-reduced diet reduces both lactose and histamine intake. It has also been reported that HIT may be accompanied by fructose malabsorption, displaying the same symptoms and caused by the same factors as LM and HIT [30]. Therefore, several questions arise: Are food intolerances (e.g., lactose or fructose malabsorption) and histamine intolerance frequently comorbid?Could disaccharide malabsorption and histamine intolerance be caused by the same primary intestine-damaging disease?And, most importantly: is HIT heavily underdiagnosed?

Interestingly, the enzymatic activity of diamine oxidase and disaccharidases in small intestine biopsies of children strongly correlate with each other: DAO and lactase (*r* = 0.80), DAO and maltase (*r* = 0.70) as well as DAO and sucrase (*r* = 0.55) [31]. Additionally, it was found that in adults increasing expiratory hydrogen in lactose intolerance is associated with additional histamine or fructose intolerance and that histamine intolerance frequently accompanies disaccharide malabsorption (HIT was present in 38% of lactose intolerant patients [32] and 36.9% of patients with carbohydrate malabsorption [30]). Nevertheless, evidence is limited and the questions stated above have to be answered in future thorough investigations of these interactions.

### 6.2. Food Consumption

Ingested food may cause abnormal histamine accumulation in several ways (Figure 2):Fish, sauerkraut, smoked meat products and some cheeses contain large amounts of histamine and may trigger HIT symptoms due to even slight overavailability of histamine [1,3,7,33,34,35]. Histamine’s presence in these foods is a result of histidine decarboxylation by microorganisms that exhibit histidine decarboxylase activity. This process occurs during food production [5]. To diagnose HIT, ingestion of histamine-rich food must be accompanied by a DAO deficit. Otherwise, histamine intoxication should be diagnosed.Some products can likely trigger histamine release. Examples are citrus fruits, papaya, strawberries, egg white, chocolate, some types of nuts, fish, and pork [1].Food rich in other biogenic amines, like tyramine or putrescine, usually accompanied by histamine, may cause competitive DAO inhibition and in combination with usually tolerable histamine amounts cause HIT [1,3]. Examples include fish, fermented sausage or sauerkraut.Putrescine most probably promotes histamine liberation from the intestinal mucosa [1]. Examples of high putrescine content food include cheese, fermented sausages, fish sauces or citrus fruits, green pepper, wheat germ and soybean sprouts [36,37].

One should also keep in mind that some foods can cause acquired DAO deficit via several mechanisms simultaneously. For example, fish is a histamine-rich, histamine-releasing and DAO-inhibitory food product. 

### 6.3. Breastfeeding

Exclusive breastfeeding for about 6 months is recommended and should be followed by the introduction of complementary foods until weaning when the child is 12 months old or older [38]. There is no direct evidence on whether histamine present in the mother’s food may trigger histamine intolerance in children. Perez et al. report that occasionally breast milk histamine concentration as well as other biogenic amines like putrescine increase when breastfeeding women are diagnosed with mastitis [39]. This molecule may act as a DAO inhibitor increasing the risk of decreased histamine metabolism [1]. To date, the earliest symptom onset of HIT was reported in a 6-months old child [7], which most probably coincided with the introduction of complementary foods [38]. Thus, the question arises whether increased concentrations of biogenic amines transferred in breast milk are high enough to cause histamine intoxication or induce histamine intolerance in an infant.

### 6.4. Drugs Administration

Certain drugs have been claimed to inhibit DAO activity and contribute to HIT pathogenesis. This list includes morphine, non-steroidal anti-inflammatory drugs, acetylcysteine, acetylsalicylic acid, clavulanic acid, isoniazid and cimetidine [1]. As many of these drugs are used in paediatric diseases [40,41], iatrogenic DAO inhibition in children should also be considered, especially if the drugs are used long term or for treating chronic conditions. However, there is no evidence on this issue yet. Nevertheless, some drugs are excluded in histamine-reduced diets for children (Table 3) [7].

## 7. Epidemiology of Histamine Intolerance in the Paediatric Population 

It is estimated that the prevalence of HIT is approximately 1% worldwide and that about 80% of those patients are adults [1]. In children, a lower diagnosis rate is possible, as children most probably do not consume as much fish, cheese or fermented sausages as adults and the symptoms may not be displayed clearly enough to diagnose HIT. Nevertheless, the true prevalence of HIT in children may be the same as in the adult population. It is also claimed that due to multifaced symptoms and many organs involved, HIT prevalence is most likely underestimated [1]. 

In one study involving children, about 8% of children who report chronic abdominal pain and have a history of histamine-rich foods consumption had decreased serum DAO concentrations [42]. Males seem to predominate in the paediatric population which is different from the adult population [7,42,43]. However, all these hypotheses need to be verified in larger cohorts. Also, the question arises about the minimum age at which HIT diagnosis can be made. As of now, a 15-month-old child is the youngest patient diagnosed with HIT [27]. However, another study claims that the symptoms of HIT were observed in a 6-month-old patient [7]. 

## 8. Histamine Intolerance Diagnosis in Children

As of now, two publications proposing a diagnostic scheme for paediatric HIT are available. Hoffman et al. propose the following diagnostic criteria: (1) patient has chronic abdominal pain; (2) patient has serum DAO concentration ≤10 IU/mL; (3) the suspected HIT symptoms improve after histamine-reduced diet; (4) positive result of histamine provocation test [42].

On the other hand, Rosell-Camps et al. recruited: (1) patients with chronic abdominal pain; (2) whose symptoms improve on a histamine-reduced diet; (3) who may have serum DAO concentration ≤10 U/mL [7]. 

The first stage of diagnosis is consistent between the studies. Gastrointestinal symptoms seem to be the most common among paediatric patients with HIT. These include chronic, diffuse abdominal pain, diarrhoea, vomiting and flatulence [7,42]. The same observation was made in the adult population, where symptoms including abdominal distension (92%), postprandial fullness (73%), diarrhoea (71%), abdominal pain (68%) and constipation (55%) were observed in the majority of the studied population [43].

However, when the second stage is considered, only about 50% of children who have decreased DAO activity report improvement in symptoms with a histamine-reduced diet [42] (Table 4). On the other hand, Rosell-Camps et al. reported that all patients with decreased DAO activity responded well to a histamine-reduced diet [7]. Therefore, it seems that the DAO activity measurement should follow the histamine-reduced diet trial, not the other way around. This scheme is also consistent with the ones proposed by Maintz et al. or Comas-Basté et al. for the adult population [1,3]. However, researchers suggest that the patient should be presenting at least two symptoms of histamine intolerance. It is unknown if this criterion was fulfilled by patients enrolled by Hoffman et al. [42], while in the study carried out by Rosell-Camps et al. all patients met this criterion [7].

As a histamine-reduced diet helps to improve symptoms in paediatric patients without DAO deficit [7], while not all patients with DAO deficit respond to a histamine-reduced diet [42], it seems that in the paediatric population decreased DAO concentration is not pathognomonic for HIT. The same issue, as well as high variations in prevalence of DAO deficit in the studied populations were observed in adults [44,45].

Moreover, Hoffman et al. claim that during a double-blind placebo-controlled trial only one out of seven patients developed symptoms of HIT and thus the test result was interpreted as positive [42]. This is consistent with another study based on the adult population. It shows that the histamine provocation test results are not reproducible and its use to diagnose HIT may be inappropriate [46]. 

In addition to these findings, it was observed that children’s serum DAO concentrations do not correlate with serum or urine histamine concentrations (R^2^ = 0.38 and 0.28, respectively) [42]. A similar outcome was observed by Rosell-Camps et al. as in eight patients who had undergone measurements only one child had urine and blood histamine concentrations above physiological level [7].

Kacik et al. implement a completely different diagnostic scheme, with patients divided into allergy and pseudoallergy groups based on their serum IgE concentration [2]. It was found that low serum DAO is associated with low total and specific IgE concentration, thus no allergy, and it was concluded that the most probable diagnosis is histamine intolerance [2]. However, the DAO concentration in the pseudoallergy group (*n* = 8) was still very high (46.40 ± 7.19 IU/mL) in comparison to ≤10 IU/mL which has been proposed as the cut-off value for the diagnosis of DAO deficit [42]. Contrary to two previous studies [7,42], the most common symptoms in the pseudoallergy group were respiratory (75%), skin (50%) and symptoms appearing after food ingestion (50%). However, the digestive system symptoms were present in only 25% of the non-allergic children, compared to 100% or nearly 100% in other studies [7,42,43]. IgE concentration, both total and specific, can be helpful in the diagnosis of allergy, but reduced DAO activity is also noticed for example in celiac disease [21]. Moreover, there is no mention if patients were treated with a histamine-reduced diet and if symptoms improved [2]. The symptoms appearing after food ingestion can be attributed to many other conditions, for example lactose intolerance [47]. Consequently, HIT diagnosis in low serum IgE patients cannot be straightforwardly based exclusively on low IgE concentration and lower DAO activity. Nevertheless, testing for total and specific IgE concentrations may be applied in the differential diagnosis between allergies and histamine intolerance.

Up till the present, no specific HIT diagnosis scheme in children or adults has been developed. Due to the diversity of symptoms [2,7,43] as well as lack of strong, unequivocal correlations between biochemical markers of HIT, HIT symptoms and HIT treatment methods HIT diagnosis remains a highly challenging diagnostic issue. 

## 9. Proposed Diagnostic Algorithm for Histamine Intolerance in Children

Based on the current evidence, instead of the flow chart method proposed by Maintz et al. [1], supported by Comas-Basté et al. [3] or the diet-provocation algorithm developed by Reese et al. [48], the authors wish to propose and discuss a new approach to HIT diagnosis based on major and minor criteria fulfilment. As the proposal derives from research involving children, a diagnostic scheme for paediatric patients may be developed.

## 10. Symptoms of Histamine Intolerance

In our proposed diagnostic scheme, exhibiting two gastrointestinal or three overall symptoms of histamine intolerance is the necessary condition for HIT diagnosis and the starting point of the diagnostic path (Table 5). Maintz et al., in their study present a different approach and suggest possible HIT diagnosis when the patient reveals at least 2 symptoms of histamine intolerance [1]. Rosell-Camps et al. noticed that paediatric patients with HIT intolerance display symptoms other than the digestive system including pruritus, rash, bronchospasm, cough, headache, nausea, and tachycardia [7]. This is consistent with the adult population, in which such symptoms, although less common than gastrointestinal ones, are not rare [43]. Also, no symptom is reported by 100% of the patients [43]. Nevertheless, an adult patient typically displays a larger number of symptoms simultaneously, ranging from 2 to 21, most frequently from 8 to12 symptoms. For children, the average number of symptoms revealed amounts to three [7]. The gastrointestinal symptoms are the most frequent and their occurrence is more indicative for HIT than symptoms from other organ systems [7,43].

At this stage of developing the diagnostic scheme, the severity and frequency of symptoms required to consider the possibility of HIT diagnosis have not yet been determined and further research is needed.

## 11. Major Criteria

Histamine-reduced diet is easy and inexpensive to administer as well as widely available, compared to experimental serum DAO measurements, colon biopsy or SNPs investigation that are expensive, invasive, require trained personnel and cannot be performed in all laboratories. Also, patients who find histamine-reduced diet improves their symptoms adhere to it after four weeks [42], which indicates well whether the therapy has been successful or not. Moreover, the diagnostic value of a histamine-reduced diet is well evidenced [3]. On the other hand, a histamine-reduced diet is not appropriate for some patients with DAO deficit [42] and it sometimes helps patients who do not have DAO deficit [7]. What is more, the ‘’histamine-reduced’’ diet tends to also reduce the intake of other substances. For example, histamine-reduced diets tend to also reduce biogenic amine consumption or at least contains reduced amounts of biogenic amines [1,3]. Nevertheless, a histamine-reduced diet appears to be the most efficient test, although biochemical confirmation of DAO deficit also provides a valuable diagnostic clue. 

DAO supplementation is also claimed to be a valuable diagnostic marker. However, there are no studies regarding its use in children, although it seems to be an effective way for treating HIT in adults [3] and thus it can be also considered as the first-step histamine-specific test for the paediatric population, too.

Interpretation of histamine oral provocation test is sometimes confusing and lacks reproducibility [42,46]. Nevertheless, despite its serious side-effects. the test can be a useful indicator of problems with histamine digestion, as it shows the direct causative effect of a large histamine dose on the patient’s body.

Some authors [3,48,49,50] also propose a 50-skin-prick test. A positive result can be indicative of HIT, however, any other possible diseases such as systemic mastocytosis or allergies have to be excluded and DAO inhibitory drugs must be discontinued before the test is performed. Thus, it can be useful when HIT is an isolated, primary disease only. With HIT as a comorbid condition, the 50-skin-prick test becomes inconclusive [50].

As a result, even the fulfilment of at least two major criteria cannot be used for a conclusive HIT diagnosis. Nevertheless, if the patient meets at least two of the criteria, HIT becomes a highly probable diagnosis. Finally, if three major criteria are fulfilled, the diagnosis of HIT can be made, as all possible histamine-specific diagnostic tests groups indicate histamine intolerance.

## 12. Minor Criteria

Some factors make HIT diagnosis even more probable but are less indicative of problems related to histamine-mediated symptoms specifically. Again, their fulfilment only increases the probability of HIT occurrence and does not allow for unequivocal diagnosis.

A food diary can help to diagnose histamine intolerance. The intake of specific food can be noted by parents\patients’ caregivers of younger patients or by adolescent patients themselves. To aid data collection younger patients may for example highlight drawings of foods they have eaten today, while for older ones a checklist can be provided. However, if it is the patient\patient’s caregiver who requests a diagnosis of HIT based on their reports, the diary’s usefulness can vary greatly, depending on its content. Some patients may report time between specific food ingestion and symptoms or provide an hour-by-hour diary, while some will just report foods eaten and symptoms experienced in a given day. On the other hand, a physician can advise the patient to complete a 24- or 48-h thorough diet-to-symptoms diary. It is crucial to provide detailed instructions and make the patient aware that the temporal relation between ingested food and symptoms is important. However, one has to remember that 24 or 48 h maybe a too short period of time and patients might not be fully compliant, leading to only partial notes and resulting in an incorrect diagnosis. Also, to date, there is no established protocol for keeping such a diary. Hence, the diagnostic value of diary data is case-dependent.

Biomarkers of histamine metabolism in urine or blood samples can also be helpful in diagnosis. However, the correlations with DAO concentration are weak [42] and their diagnostic usefulness is still being investigated [3,51]. Moreover, it is difficult to provide clear guidance on when tests to exclude allergies or mastocytosis should be made, as it is indicated in the algorithm of Maintz et al. [1]. 

## 13. Comorbidities

In the light of the proposed diagnostic criteria, it is worth emphasizing that HIT may be, and in most cases probably is, a comorbidity. This is supported by studies correlating DAO/HNMT deficit (measured by SNPs) or increases in blood serum histamine concentrations with the severity of allergic diseases like atopic dermatitis, allergic rhinitis or asthma and bronchial hyperresponsiveness [15,16,18,19]. Additionally, DAO deficits were observed in bowel diseases, for example, celiac disease or acute gastroenteritis in children [21,22] and recurrent urticaria, colon adenoma, carbohydrate malabsorption as well as food allergies in adults [30,32,52,53,54,55,56]. Also, the DAO activity level was proposed as a bowel integrity marker [20,21]. As these are allergic and gastrointestinal diseases, the frequency or severity of histamine-mediated symptoms increases most likely due to the elevation of serum histamine concentration, leading to possible co-occurrence of histamine intolerance. Such possibility poses an additional diagnostic challenge and was included as the last additional criterion in the proposed HIT diagnosis scheme (Table 5).

## 14. Treatment of Histamine Intolerance

According to recent studies, a histamine-reduced diet is a treatment of choice for HIT (Figure 3, [7,42]). However, its effectiveness in symptom improvement in children varies. This choice of treatment is consistent with studies on adults with HIT [3]. Similarly, the effectiveness varies, but in general, the response is rather good [3]. Additionally, Rosell-Camps et al. report successful treatment of more severe cases of HIT in children with H_1_ and H_2_ antihistamines as well as oral zinc supplementation as a combination therapy along with a histamine-reduced diet [7].

For adults, DAO supplementation is also proposed and the results are promising [1,3]. However, there is no study on a paediatric population that evaluates this treatment method. Some companies offer DAO supplements [57], but currently it is not possible to provide patients with evidence-based recommendation on this topic. 

During cyclophosphamide administration in neuroblastoma paediatric patients, the pre-administration of dietary fibre inhibited the influence of cyclophosphamide on DAO activity [20], which shows a potential treatment option of drug-induced HIT. Also, it might be applied in other bowel-disease-induced DAO deficits [53,54,55]. Nevertheless, this therapy needs to be investigated more thoroughly to enable its assessment and drawing reliable conclusions.

## 15. Conclusions

Histamine intolerance has to be considered in children with non-specific gastrointestinal complaints and histamine related symptoms. Although decreased DAO activity seems to be the most relevant finding in patients with HIT, its significance as a diagnostic marker needs to be verified in large prospective cohort studies. A histamine-reduced diet is a treatment of choice for HIT, but other treatment options like DAO or dietary fibre supplementation might be suggested to patients who do not respond to an elimination diet. Secondary HIT accompanying allergies and bowel pathologies could be responsible for worsening the symptoms of these conditions.

## Figures and Tables

**Figure 1 nutrients-13-01486-f001:**
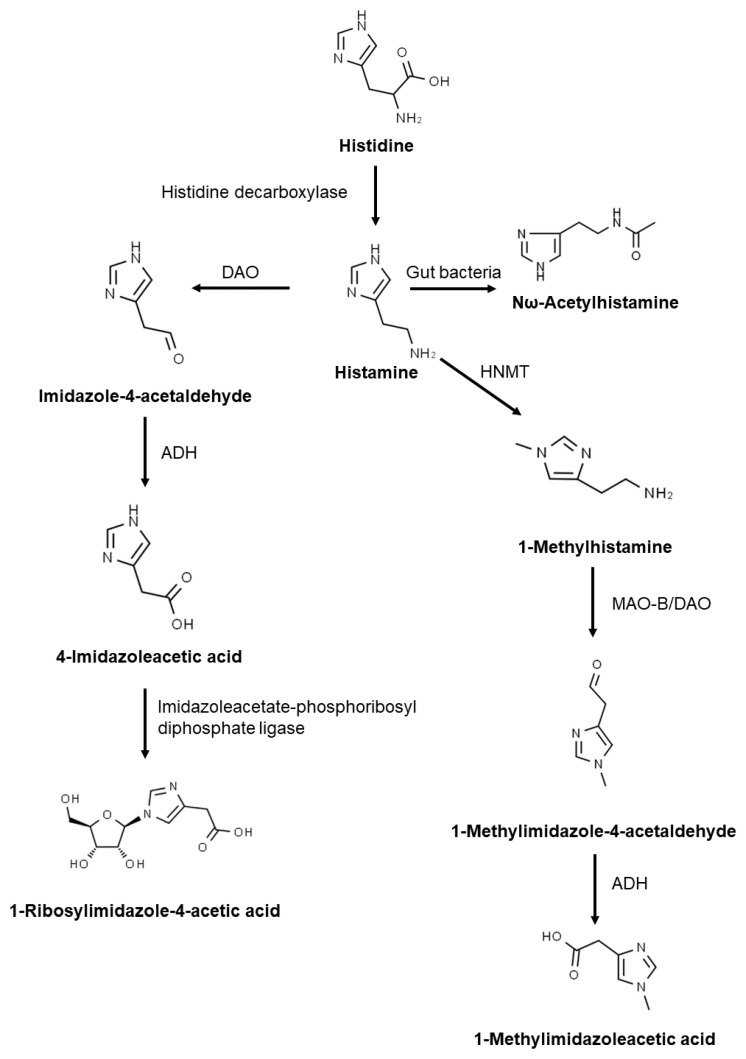
Histamine metabolism. DAO: diamine oxidase; MAO: monoamine oxidase B; HNMT: histamine-N-methyltransferase; ADH: aldehyde dehydrogenase.

**Figure 2 nutrients-13-01486-f002:**
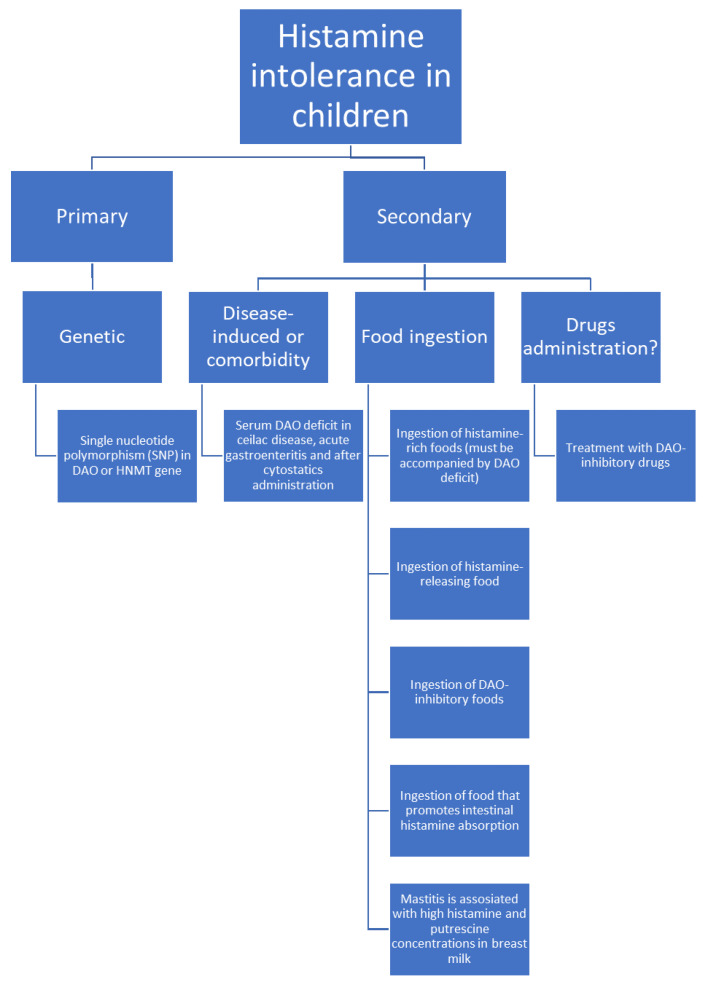
Possible histamine intolerance aetiologies in children.

**Figure 3 nutrients-13-01486-f003:**
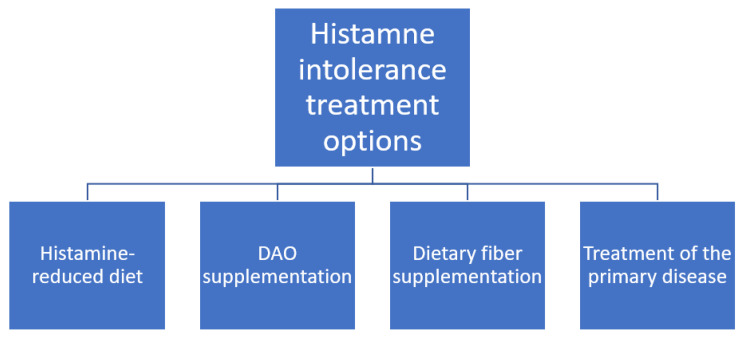
Histamine intolerance treatment options.

**Table 1 nutrients-13-01486-t001:** Food allowed on a histamine-reduced diet [1,7,8,9].

Food	Histamine Content: Mean [mg/kg]	Histamine Content: Range [mg/kg]
Water	-	-
Coffee	-	-
Tea	-	-
Bread	ND	-
Cereals	ND	-
Oats	ND	-
Pasta	ND	-
Rice	ND	-
Yoghurt	ND	-
Asparagus	0.34	ND–1.42
Beans	ND	-
Broccoli	ND	-
Carrot	ND	-
Cauliflower	ND	-
Courgette	ND	-
Cucumber	ND	-
Lettuce	ND	-
Mushroom	ND	-
Onion	ND	-
Potatoes	ND	-
Pumpkin	0.28	ND–1.90
Apple	ND	-
Banana	ND	-
Cherry	ND	-
Peach	ND	-
Pineapple	ND	-
Plum	ND	-
Grapefruit juice	0.31	ND–1.74
Orange juice	0.46	ND–1.32
Pineapple juice	2.44	ND–4.61
Herbs and spices	ND	-
Pepper	ND	-
Vegetable oil	-	-
Vinegar	-	-
Olives	-	-
Fresh meat	ND	-
Cooked meat	0.3	ND–4.8
Poultry	3	-
Mozzarella	ND	-
Cream cheese	3	-

ND: not detected; -: no measurement.

**Table 2 nutrients-13-01486-t002:** Food to be avoided on a histamine-reduced diet [1,7,8,9].

Food	Histamine Content: Mean [mg/kg]	Histamine Content: Range [mg/kg]
Cod (fresh)	-	2–77
Tuna (frozen)	ND	-
Tuna (smoked or salted)	ND	-
Tuna (canned)	0.33	1–402
Mackerel (frozen)	-	1–20
Mackerel (smoked or salted)	-	1–1788
Mackerel (canned)	-	ND–210
Herring (frozen)	-	od 1 do 4
Herring (smoked or salted)	-	5–121
Herring (canned)	-	1–479
Sardine (frozen)	ND	-
Sardine (smoked or salted)	-	14–150
Sardine (canned)	-	3–2000
Shellfish/Shrimp paste	328	-
Fish sauce	574.7	-
Parma ham	1100	-
Sobrassada/Soppresata	21.9	-
Saucisson	71	-
Salami	-	1–654
Fermented ham	-	38–271
Fermented sousage	21.9	ND–650
Parmesan (cheese)	40.64	10–581
Blue cheese	376.6	-
Gouda (cheese)	-	10–900
Emmental (cheese)	-	5–2500
Roquefort (cheese)	9.9	-
Camembert (cheese)	-	0–1000
Cheddar (cheese)	-	0–2100
Sauerkraut/fermented cabbage	37	0–229
Soy derivatives	307	ND–307
Chocolate	0.58	-
Vanilla	-	-
Citrus	-	-
Kiwi	ND	-
Nuts	0.45	ND–11.86
Strawberry (dry)	-	-
Pineapple (dry)	-	-
Papaya (dry)	-	-
Spinach	31.77	9.46–69.71
Tomato (fresh and sauce)	22	ND–22
Eggplant	39.42	4.17–100.6
Eggs	-	-

ND: not detected; -: no measurement.

**Table 3 nutrients-13-01486-t003:** Drugs mediating histamine intolerance and excluded on histamine-reduced diet in children, adopted from Rosell-Camps et al. [7].

Type of Drug/Substance	Example
Contrast media	-
Muscle relaxants	Pancurorium, alcuronium, D-tubocurarine
Anaesthetics	Thiopental
Analgesics	Morphine, pethidine, NSAIDs, ASA, metamizole
Local anaesthetics	Prilocaine
Cardiotonics	Dobutamine, dopamine
Antihypertensives	Verapamil, alprenolol, dihydrazine
Antiarrhythmics	Propafenone
Diuretics	Amiloride
Antibiotics	Cefuroxime, isoniazid, pentamidine, clavulanate, chloroquine
Mucolytics	Ambroxol, acetylcysteine
Bronchodilators	Aminophylline
Cytostatics	Cyclophosphamide
Antidepressants	Amitriptyline
Prokinetics	Metoclopramide
Antihistamines	Cimetidine

**Table 4 nutrients-13-01486-t004:** Comparison of recent studies on histamine intolerance in children.

Clinical Outcome	Number of Patients	
	Rosell-Camps et al. [7]	Hoffman et al. [42]
Abdominal pain	16 (100%)	31 (100%)
DAO deficit	14 (87.5%)	31 (100%)
Males	11 (68.8%)	17 (54.8%)
DAO deficit and positive response to histamine-reduced diet	14 (100%)	16 (43.2%)
High serum histamine level	1 out of 8 measured (12.5%)	22 (71.0%)
High urine histamine level	1 out of 8 measured (12.5%)	13 (41.2%)

**Table 5 nutrients-13-01486-t005:** Proposed algorithm for histamine intolerance diagnosis in children.

**Must-Have Criteria**
Presenting ≥ 3 symptoms of histamine intolerance in total or ≥2 gastrointestinal symptoms
**≥2 Major criteria**
Positive histamine oral provocation test
OR
positive 50-skin-prick test when other possible pathologies giving positive result are excluded
serum DAO deficit (≤10 IU/mL)
OR
symptoms improvement after DAO supplementation (4–8 weeks)
OR
symptoms improvement after DAO inhibitory drugs dismission
OR
identification of single nucleotide polymorphism responsible for DAO/HNMT deficit
OR
decreased DAO activity in colon biopsy
Symptoms improvement after histamine-reduced diet (4–8 weeks)
**Additional criteria that increase the probability of histamine intolerance diagnosis**
Correlation between specific food
consumption and symptomatology based on a diet diary
High concentration biomarkers of histamine metabolism in urine, blood or stool samples
Exclusion of mastocytosis (tryptase)
Exclusion of food allergies (skin prick test)
Exclusion or confirmation of other underlying diseases relevant to presented symptoms

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
