# Peer review of "Histamine Intolerance in Children: A Narrative Review"

_nutrients, 2021, doi:10.3390/nu13051486_

Round 1

Reviewer 1 Report

This review aims to decribe histamine intolerance in children. Generally, a review on this topic is necessary. But this manuscript must be rewritten, structured and the English needs thorough correction by a native speaker.

Here are some examples of necessary corrections:

At the place of "Aims" should be an "Introduction" with the description of background and necessity of this review. Aims is the last line of this introduction. 

It is impossible to eat  "histamine-free". This must be changed to histamine-reduced diet throughout the manuscript.

Table 1 needs to be split (into three tables) and clearly adapted to the needs of children. I wonder why alcohol (in children?) is listed. An example would be: foods allowed for children, foods not allowed and drugs used and mediating HIT in children.

If a diagnostic algorithm needs to be proposed it should be for children, only.

Diseases causing low diamine oxidase and/or histamine intolerance in children must be desribed in detail.

Comorbidities like food intolerances in children need to be included.

References e.g. reference 7: is the journal stile Inflammation Research? or is Inflamm. Res. correct? This needs correction throughout the references.

References are usually a list of scientific publications from mainly peer reviewed journals, only. References like 45 to 47 are not scitable sources.

Author Response

Dear Reviewer, thank you for valuable comments regarding our manuscript. We agree with all your suggestions. You can find the detailed response below (annotated as “###”)

This review aims to decribe histamine intolerance in children. Generally, a review on this topic is necessary. But this manuscript must be rewritten, structured and the English needs thorough correction by a native speaker.

### English was corrected by two independent certified medical proofreaders

Here are some examples of necessary corrections:

At the place of "Aims" should be an "Introduction" with the description of background and necessity of this review. Aims is the last line of this introduction.

### Introduction was added:

Introduction

Histamine intolerance, HIT, is defined as a disequilibrium of accumulated histamine and the capacity for histamine degradation [1]. This clinical term addresses a non-immunologically mediated pathology when histamine ingested with food is not particularly high, but its degradation is decreased [1,2]. Thus, some individuals are unable to degrade ingested histamine properly, which subsequently causes sensitivity to normal or even low histamine levels in food. Research on this topic has significantly accelerated during the last decade and the awareness of histamine intolerance has been growing among researchers and in the general public [3]. However, to date, no review on HIT in children has been published. This paper aims is to provide a review on etiopathology, epidemiology, possible diagnostic algorithms as well as diagnostic challenges of histamine intolerance in children. The differences between the paediatric and adult populations are also highlighted.

It is impossible to eat  "histamine-free". This must be changed to histamine-reduced diet throughout the manuscript.

### we changed to histamine-reduced nomenclature

Table 1 needs to be split (into three tables) and clearly adapted to the needs of children. I wonder why alcohol (in children?) is listed. An example would be: foods allowed for children, foods not allowed and drugs used and mediating HIT in children.

### we added tables with foods allowed and foods to be avoided on histamine-reduced diet (Table 1 and 2, respectively).

Moreover, we deleted description of alcoholic beverages from the text.

Also, the drugs most probably causing HIT were moved to the Table 3.

If a diagnostic algorithm needs to be proposed it should be for children, only.

### We added the following description:

As the proposal derives from research involving children, a diagnostic scheme for paediatric patients may be developed.

Diseases causing low diamine oxidase and/or histamine intolerance in children must be desribed in detail. Comorbidities like food intolerances in children need to be included.

### We included the comorbidities as well as diseases that causes DAO deficit in the following description:

Disease-induced histamine intolerance

DAO deficit was found in children undergoing cyclophosphamide administration in neuroblastoma paediatric patients [20], in celiac disease [21], in acute gastroenteritis [22] and during protracted diarrhoea [23]. In adults, decreased DAO excretion was reported in other diseases including chronic urticaria, viral hepatitis [1] and inflammatory bowel diseases [3]. These pathologies also occur in childhood [24–26] but insofar no evidence of DAO deficit in those conditions has been reported in children, although, as one report describes, histamine intolerance was diagnosed in a child with viral intestinal infection [27].

Thus, it appears like DAO deficiency may be caused by various factors such as the presence of a primary disease or cytostatic treatment. The common denominator is intestine damage. Similar primary factors cause secondary lactose deficiency described as reduced lactase activity [28]. Lactase, an enzyme responsible for lactose digestion, is produced, similarly to DAO, by intestinal cells. Also similarly to DAO deficiency, symptoms of lactose malabsorption (LM) include flatulence, diarrhoea, abdominal pain as well as systemic symptoms like headaches [29,30]. Morover, the exclusion of the same dietary products may lead to an improvement in symptoms. For example, a cheese-reduced diet reduces both lactose and histamine intake. It has also been reported that HIT may be accompanied by fructose malabsorption, displaying the same symptoms and caused by the same factors as LM and HIT [30]. Therefore, several questions arise:

  • Are food intolerances (e.g. lactose or fructose malabsorption) and histamine intolerance frequently comorbid?
  • Could disaccharide malabsorption and histamine intolerance be caused by the same primary intestine-damaging disease?
  • And, most importantly: is HIT heavily underdiagnosed?

Interestingly, enzymatic activity of diamine oxidase and disaccharidases in small intestine biopsies of children strongly correlate with each other: DAO and lactase (r = 0.80), DAO and maltase (r = 0.70) as well as DAO and sucrase (r = 0.55) [31]. Additionally, it was found that in adults increasing expiratory hydrogen in lactose intolerance is associated with additional histamine or fructose intolerance and that histamine intolerance frequently accompanies disaccharide malabsorption (HIT was present in 38% of lactose intolerant patients [32] and 36.9% of patients with carbohydrate malabsorption [30]). Nevertheless, evidence is limited and the questions stated above have to be answered in future thorough investigations of these interactions.

References e.g. reference 7: is the journal stile Inflammation Research? or is Inflamm. Res. correct? This needs correction throughout the references.

### We corrected this issue.

References are usually a list of scientific publications from mainly peer reviewed journals, only. References like 45 to 47 are not scitable sources.

### We deleted fragments with these references. Moreover, 13 new sources were cited throughout the text.

Reviewer 2 Report

The review entitle “Histamine intolerance in children” aims to update the etiology, epidemiology and diagnosis of histamine intolerance in children. The manuscript as a review is a bit short, especially when compared to recent reviews as reference [3] (Comas-Basté et al., 2020). However, he proposes a new algorithm for the diagnosis of histamine intolerance that may be of interest. In my opinion, it should not be considered a review, and the manuscript needs a thorough revision to be considered for publication.

The different levels of histamine in food is important for distinction between histamine intolerance and histamine intoxication, it should be included which levels are considered high. The description of the metabolic degradation of histamine is poor. Its degradation pathways are important in histamine intolerance.

English language and style should be revised.

Author Response

Dear Reviewer, thank you for valuable comments regarding our manuscript. We agree with all your suggestions. You can find the detailed response below (annotated as “###”)

The review entitle “Histamine intolerance in children” aims to update the etiology, epidemiology and diagnosis of histamine intolerance in children. The manuscript as a review is a bit short, especially when compared to recent reviews as reference [3] (Comas-Basté et al., 2020). However, he proposes a new algorithm for the diagnosis of histamine intolerance that may be of interest. In my opinion, it should not be considered a review, and the manuscript needs a thorough revision to be considered for publication.

The different levels of histamine in food is important for distinction between histamine intolerance and histamine intoxication, it should be included which levels are considered high.

### we added a table with reference values for histamine content in different type of foods (Table 1 and 2). Moreover, we discussed this issue in the text:

Histamine intolerance or histamine intoxication?

In HIT, histamine accumulation in intestinal mucosa and blood plasma is caused by decreased histamine degradation, resulting most commonly from low DAO activity. Additionally, tissue HNMT is inhibited by histamine metabolites [7]. Thus, HIT can occur even when food with moderate or even low amounts of histamine is ingested.

This should be differentiated from the situation when individuals presenting with normal histamine degradation enzyme activity ingest large quantities of histamine-rich food (mainly fish) and develop histamine-induced symptoms, which is classified as histamine intoxication [3].

Histamine acting on its H1 - H4 receptors [10] is a common denominator in both pathologies, which results in the appearance of histamine-mediated symptoms [3].

The recent meta-analysis shows that the mean histamine content in foods causing histamine intoxication is about 1100 mg/kg (95% confidence interval 422-2900 mg/kg) and almost all intoxications (98%) result from fish consumption [11]. Legal limits for histamine content in food products (mainly fish) are set, in the majority, at between 100 and 400 mg/kg [12]. Thus, the histamine content of food causing intoxication is substantively higher than stated in food safety regulatory guidelines.

For comparison, it appears like foods that can be allowed on a histamine-reduced diet contain less than 5mg/kg of histamine (Table 2). On the other hand, most recommendations for histamine-reduced diet list foods with a wide range of histamine content, including some foods which may contain histamine concentrations near the legal limits for histamine content in foods. (Table 2) [12].

The description of the metabolic degradation of histamine is poor. Its degradation pathways are important in histamine intolerance.

### We added a paragraph about histamine degradation pathways and a figure that illustrates it (figure 1):

Histamine metabolism

Histidine decarboxylase catalyses the production of histamine from histidine (Figure 1). Histamine is metabolised by two primary enzymes: diamino oxidase, DAO, and histamine-N-methyltransferase, HNMT [3,4]. HNMT is responsible for intracellular histamine metabolism, while DAO is a secretory protein that metabolises histamine extracellularly [5]. DAO has a higher expression than HNMT [3], being the primary barrier for intestinal histamine absorption [5].

Imidazole acetaldehyde produced by DAO is subsequently metabolised by aldehyde dehydrogenase, ADH, to imidazole acetic acid. Acetic acid is then degraded into 1-ribosylimidazole-4-acetic acid. In the second metabolic pathway, N-methylhistamine produced by HNMT is oxidised by monoamine oxidase B leading to N-methyl imidazole acetaldehyde that is further transformed into methyl-imidazole acetic acid by ADH. Histamine can also be metabolised to Nω-acetylhistamine by gut bacteria [4,6].

English language and style should be revised.

### English was corrected by two independent certified medical proofreaders

Round 2

Reviewer 2 Report

The main concerns raised in the previous version have been solved.

Minor comments:

Review the edition of the manuscript. Prevent isolated section headings at the end of a page, and the split of table 5 between two pages. Check out word split in figures 2 and 3.

Author Response

Dear Reviewer,

we made all the necessary corrections.

Thank you for all your advices.